# Spatial Accessibility of Primary Care in the Dual Public–Private Health System in Rural Areas, Malaysia

**DOI:** 10.3390/ijerph20043147

**Published:** 2023-02-10

**Authors:** Jabrullah Ab Hamid, Muhamad Hanafiah Juni, Rosliza Abdul Manaf, Sharifah Norkhadijah Syed Ismail, Poh Ying Lim

**Affiliations:** 1Department of Community Health, Faculty of Medicine and Health Sciences, Universiti Putra Malaysia (UPM), Serdang 43400, Selangor, Malaysia; 2Institute for Health Systems Research, National Institutes of Health, Ministry of Health Malaysia, Blok B2, Kompleks NIH, No. 1, Jalan Setia Murni U13/52, Seksyen U13 Setia Alam, Shah Alam 40170, Selangor, Malaysia; 3Department of Environmental and Occupational Health, Faculty of Medicine and Health Sciences, Universiti Putra Malaysia (UPM), Serdang 43400, Selangor, Malaysia; 4Research Institute on Ageing (MyAgeing), Universiti Putra Malaysia, Serdang 43400, Selangor, Malaysia

**Keywords:** spatial accessibility, primary care, rural, floating catchment area method, factor

## Abstract

Disparities in access to health services in rural areas represent a global health issue. Various external factors contribute to these disparities and each root requires specific remedial action to alleviate the issue. This study elucidates an approach to assessing the spatial accessibility of primary care, considering Malaysia’s dual public–private system specifically in rural areas, and identifies its associated ecological factors. Spatial accessibility was calculated using the Enhance 2-Step Floating Catchment Area (E2SFCA) method, modified as per local context. Data were secondary sourced from Population and Housing Census data and administrative datasets pertaining to health facilities and road network. The spatial pattern of the E2SFCA scores were depicted using Hot spot Analysis. Hierarchical multiple linear regression and geographical weight regression were performed to identify factors that affect E2SFCA scores. Hot spot areas revolved near the urban agglomeration, largely contributed by the private sector. Distance to urban areas, road density, population density dependency ratios and ethnic composition were among the associated factors. Accurate conceptualization and comprehensive assessment of accessibility are crucial for evidence-based decision making by the policymakers and health authorities in identifying areas that need attention for a more specific and localized planning and development.

## 1. Introduction

Disparities in access to healthcare services represents a known global health issue that is even more pronounced in low- and middle-income countries [1]. Access of healthcare defined as the relative ease with which health services can be utilized by the population [1,2]. Rural populations often experience a lack of access to healthcare services compared to their urban counterparts [3,4], largely constrained by travel impedance and availability of services [5,6]. Similar circumstances exist in Malaysia, where less developed areas generally have lower access to healthcare in terms of provider-to-population ratios [7], especially in the private sector which is mainly concentrated in urban areas [8,9]. Nonetheless, healthcare services and population demand are hardly to be equally distributed [5]. The geographic disparities in healthcare access existed even within rural areas, where lower access is often associated with rural areas located further from the nearest urban area, especially due to unmatched population and service distribution [3,10,11,12].

Access to health care can be defined in several ways depending on the components of access that being measured [13]. Several theoretical frameworks for healthcare accessibility exist [1,14,15,16,17], dividing accessibility into spatial and non-spatial components [16]. Spatial accessibility is one dimension of access, encompasses of geographic accessibility (physical distance) and availability (adequacy of services at specified location) [1]. Spatial accessibility can be measured using two-step floating catchment area (2SFCA), consolidating the elements of cross-boundary interaction, proximity, service availability, service supply, population demand and distance decay into one integrated measure [18,19]. It involves two-step process considering the: (i) service catchment area; and (ii) population catchment area. This method has been continually improved since the last decade and various versions of 2SFCA exist to suit specific local contexts and scenarios [20,21,22,23]. Currently, the E2SFCA method (and its variant) has been widely used to measure spatial accessibility in primary care services in both urban and rural settings [3,11,19,21,24,25,26,27], as well as for other healthcare services. The advantages of the E2SFCA over 2SFCA method were well documented in the literatures, where 2SFCA tends to overestimate as it does not include the distance decay function [11,21,28]. Comparison between the two methods was extensively described elsewhere [10,21,29] and currently the E2SFCA is considered as the standard method replacing the original 2SFCA [30]. Many of the studies using the E2SFCA method have been conducted in high-income and developed countries [3,11,19,21,23,24,26,31,32], but there have been a growing number of studies conducted in low- and middle-income countries as well, primarily in China [33,34] and India [27,35], which stand to benefit from the technology advancement of geographical information systems (GIS) and the availability of data. The E2SFCA method has proven to be an acceptable measure of healthcare accessibility as it specifically identifies both low- and high-access areas for practical healthcare planning and allocation purposes for policymakers.

In the existing healthcare access frameworks also acknowledge the role of external factors affecting the accessibility in general [1,15,16,17]. Those external factors can be categorized as geographical (spatial) and demographic (non-spatial) factors. Geographical factors generally includes specific administrative regions, remoteness and distance of that particular area to the main city [12,28]. Demographic factors includes population characteristics such as density, composition and socio-economic status as well as the environment or general infrastructure attributes of that particular area [12,22,36]. Different factors may require different remedial action by the health authorities or local government to specifically address the issue locally.

Ensuring adequate access to quality essential health services has always been an integral part of the nation’s progress towards universal health coverage as stated in the Sustainable Development Goal 3.8. This requires continuous monitoring, prompt and accurate measure of population access to health care. Despite its growing popularity of the E2SFCA method in measuring the level of access to healthcare, there have been limited studies in Malaysia. Therefore, it is crucial to advance the current understanding of accessibility to primary care services by using the E2SFCA method in the local context of Malaysia, which has not been carried out before. Such findings could provide baseline data for future studies as well as for formulation of policies related to healthcare accessibility.

This could elucidate the following: (i) what is the approach to assess the of spatial accessibility of primary care considering Malaysia’s dual public–private system specifically in rural areas; (ii) what are the differences in spatial accessibility between public and private primary care; and (iii) what are the factors affecting the spatial accessibility. By adapting the E2SFCA method, this study will respond to questions concerning the spatial accessibility of primary care among rural populations in Malaysia. Thus, this ecological study aimed to determine the level, depict the pattern of the spatial accessibility (calculated using a modified E2SFCA method) of primary care services in rural areas in Selangor. This study also seeks to identify ecological factors that associated to the spatial accessibility scores and how does it varied across the study area using both hierarchical multiple linear regression and geographical weight regression.

## 2. Materials and Methods

### 2.1. Study Area

Selangor state is located in the west-central region of Peninsular Malaysia, encircling two federal administrative centers (Kuala Lumpur and Putrajaya) (Figure 1a). The state is comprised of nine administrative districts, namely Gombak (GBK), Hulu Langat (HUL), Hulu Selangor (HUS), Klang (KLG), Kuala Langat (KUL), Kuala Selangor (KUS), Petaling (PET), Sabak Bernam (SBK), and Sepang (SPG). Although Selangor state has one of the highest urbanization rates in Malaysia at 91.4% [37] and generally has higher socioeconomic indicators compared to other states [38], rural Selangor served as a good location for this case study to assess the E2SFCA method since: (1) rural Selangor has the lowest percentage of households within 5 km of public health facilities coverage in Peninsular Malaysia [38]; and (2) Selangor, being the most populated state, has a rather low public health facility to population ratio of 1:32,555, which is about three times lower than the national average [39]. Thus, a possibility of overburden of public facilities was noted, as well as disparities in access that could eventually exist in rural Selangor. Moreover, Selangor contains the urban areas of Kuala Lumpur and its surrounding cities, and large variations and differences in spatial accessibility of primary care could potentially be observed due to the broad range of geographic characteristics, from deep rural areas to suburban areas near the core of the large urban center.

### 2.2. Data Sources

Three secondary datasets were obtained from several government agencies including population data, primary care facilities data, and road network data. All data from Selangor state (including urban and rural areas) and from adjacent states were obtained to consider population demand and service distribution, accounting for cross-boundary interaction.

Population data were sourced from the 2010 Population and Housing Census provided by DOSM (Figure 1a), aggregated at enumeration block (EB) level, an imaginary boundary formed for census purposes. All 1349 rural EBs were included in this study, classified as non-gazetted area with population less than 10,000 [38]. PET district is a 100% comprised of urban EBs, thus excluded in this study. The Department of Statistics Malaysia (DOSM) defines a rural area as a non-gazetted area with a population of less than 10,000 [38]. Each EB contains of about 100 living quarters and averaged at 345 individuals per EB. Other population characteristics extracted were age groups, sex, and ethnicity.

Data on both public and private primary care facilities (clinics) were obtained from the Malaysian Ministry of Health (MOH), based on year 2017 (Figure 1b). Public clinics refers to one that managed by MOH (health clinic and mobile service), operated by at least one doctor. Private clinics refers to general practitioners that specifically provide modern medicine. Coordinates of these facilities were geocoded based on their street addresses. Due to the unavailability of data for the number of doctors per clinic, estimated numbers depending on the clinic types were used. It was estimated that the median number of doctors for public clinic was six, whereas the median number of doctors for both public mobile services and private clinics in Selangor was one [40,41]. Data on operating hours was used to estimate each clinic’s availability.

Data on road networks (Figure 1b) were obtained from the National Geospatial Centre Malaysia. Due to the unavailability of data on the actual speed limit for each road, speed limits were estimated based on the type of the road: 90 km/h for expressways, 60 km/h for federal and state roads, and 30 km/h for residential roads [42]. All data includes of Selangor state (including urban and rural) and its neighboring to cater for cross-boundary interaction of the population demand and service distribution.

### 2.3. Enhanced Two-Step Floating Catchment Area (E2SFCA)

This study adopts the E2SFCA model by Luo and Qi [21] with several consideration and modification to suit our national context of the primary care setting, which will be explained in the following paragraphs. Population data were aggregated at the smallest spatial aggregation available in the census data (i.e., EB). Considering the dual public–private primary care system, access scores for public (*Aspub*) and private (*Aspri*) providers were calculated separately and then summed to produce a total access score (*Astot*). The term “E2SFCA scores” in this document will refer to the all three *Aspub* score, the *Aspri* score, and the *Astot* score.

Distances separation between each population point (EB) and clinic were based on travel time with motorized vehicles (as the mode of transport) via the road network and were calculated using the ‘closest facility’ function of the ArcGIS (ESRI, Redlands, CA, USA, version 10.7). The travel time was calculated based on the length and speed limit of the road along the route between the two-point location, with private motorized vehicle as the mode of transport this is the commonest mode [43]. Population point was based on weighted centroid using road density as a proxy measure of the point location where the population concentrated at within the EB. This will allow for better precision in calculating the distance of separation [44], knowing that population settlements concentrated and located with close proximity to road [45,46].

Another consideration for the local context is the catchment size. A catchment size of 30 min (travel time) was used in this study (rather than the commonly used 60 min specifically for rural areas) [10,19,22]. This decision was grounded based on the national survey where mean travel time (self-reported) for rural populations seeking outpatient care was 30 min [43]. The number of clinics to be identified within 30 min of population’s catchment size were capped at nearest 100, as to mitigate the redundancy effect due to too many clinics [47]. A three-step zonal distance decay function—with fast decay weights of 0.945, 0.400, and 0.010 for the 0–10, 11–20, and 21–30 min zones, respectively [48]—was used in the calculation as it produces a sharper decay effect with more distinguishable reduced access scores, as compared to a slow decay [20].

Modification made to the original formula are the incorporation of the clinic availability (service supply) and health needs (population demand) as additional weight modifiers into the E2SFCA calculation in step 1, which is the initial supply–demand ratio of each clinic. Clinic availability (*CA*) was quantified based on estimated total operating hours per week for each clinic, transformed to a range of 0.5–2.0, where 24-h clinic is assigned weight of 2.0, clinic that operates from 8 am–10 pm as 1.5, clinics that operates at common office hour (8 am–5 pm) on weekdays weighted as 1.0 and clinics that operates less than that as 0.5. Health need (*HN*) was quantified as the percentage of the total population of the EB that could be considered vulnerable (toddlers aged <5 years, elders aged >64, and women aged 15–45) [19]. High *HN* denotes EBs with a higher percentage of vulnerable people. HN was then transformed to a range of 0.5–2.0 to be incorporated in step 1 of the E2SFCA calculation [49]. Therefore, the number of doctors (physicians), weighted by its operating hours to indicate total supply for each clinic, whereas the total demand was the total population, weighted by the health need. In contrast to the original formula, the supply-demand concept was without the two additional weight modifiers (*CA* and *HN*). With that, the modified formula of the E2SFCA calculation were as below.

Step 1—Assigning an initial supply–demand ratio (*R_j_*) to each service location, by determining the population within the catchment area of service:(1)$$R_{j}=\frac{S_{j}\times CA_{j}}{\sum_{k\in \left(D_{jk}\in D_{r}\right)}P_{k}W_{r}\times HN_{k}}$$

Step 2—Summation of *R_j_* within each population location (EB), to get final spatial access scores:(2)$$A_{k}^{F}=\overset{}{\underset{j\in \left(D_{jk}\in D_{r}\right)}{\sum }}R_{j}W_{r}$$
where *R_j_* is the supply–demand ratio within the catchment area for clinic location *j*; *S_j_* is the total number of doctors (supply) for clinic location *j*; *CA_j_* is the clinic availability weight for clinic location *j*; *P_k_* is the total population at EB location *k*; *D_jk_* is the travel time between *j* and *k*; *D_r_* is the *r*th zone (*r* = 1–3); *W_r_* is the distance weight for the *r*th travel time zone; *HN_k_* is the health need weight for the population at EB location *k*; and *A^F^_k_* is the final accessibility E2SFCA score at EB location *k*. More details on the E2SFCA calculation have been mentioned in previous studies [19,21,22]. The final calculated E2SFCA scores were then multiplied by 1000 to ease numerical presentation, which can be interpreted as ratio of one doctor per 1000 population within 30 min catchment size.

### 2.4. Spatial Pattern and Spatial Statistics

Choropleth mapping was used to visualize the spatial pattern of the calculated E2SFCA scores. The E2SFCA scores were ranked and grouped into five classes based on Jenks natural breaks classification, which is a standard method of classification for visualization in GIS applications [50]. Spatial autocorrelation analysis using Global Moran’s I statistic [51] was performed to ensure that the observed spatial pattern was not due to random arrangement. To further investigate where the EBs with high or low E2SFCA values were clustered, Hot Spot Analysis using Getis-Ord Gi* statistics was performed to indicate high/low-value areas with significant confidence interval (CI) levels of 90%, 95%, and 99% [52]. Positive Gi values indicate hot spots (EBs with high E2SFCA scores clustered together), and negative Gi values indicate cold spots (EB with low E2SFCA scores clustered together). All maps were generated by the authors and all spatial statistical tests were performed using ArcGIS Desktop 10.7 (ESRI, Redland, CA, USA).

### 2.5. Assessing Ecological Factors That Associates to the Spatial Accessibility

#### 2.5.1. Theoretical Approach and Studied Variables

In this study, factors that were associated with the health care access were based on previous frameworks that have been published where population characteristics is one of the main component that affects accessibility in general [1,15,16]. This study included variables that were available from the Census 2010 data for extraction. This study uses the term ‘ecological’ factors as the variables in the data were the attributes of a population, aggregated at the EB level. The ecological factors were divided into two main categories: (i) geographical—district, locality and distance to nearest urban area; and (ii) demographic—population density, road density, household size, percentage of certain population (female, vulnerable, Malay ethnicity and marginalized) in the EB, and the dependency ratios. Definition for the explanatory variables used in this study are presented in Table 1. Dependent variables in this study are the calculated E2SFCA scores.

#### 2.5.2. Hierarchical Regression and Geographical Weight Regression

Ecological factors associated to the spatial accessibility (E2SFCA) scores were modelled separately, using the hierarchical linear regression for accessibility score public, private and total (hereafter *Aspub*, *Aspri*, and *Astot* score) respectively. Two separate blocks of explanatory variables were divided into (i) geographical; and (ii) demographic factors. Block 1 consists of administrative district, locality and distance to nearest urban area. Block 2 consists of population characteristics such as density, average household size, percentage of female population and etc. as described in Table 1. This sequential order of entry was based on a priori hypothesis in which the additional variance may be explained by demographic of the population after accounting for the variance related to geographical factors, which contributes the most [53]. Upon entry of each block, the adjusted R^2^ change was assessed to determine the amount of variance (%) explained by the models. Model assumptions for linear regression were checked accordingly and explanatory variables retained (variables selection stage) in the final model were based on lower Akaike’s information criterion (AIC). Additionally, a geographical weight regression (GWR) was performed to explore the spatial heterogeneity of each explanatory variables, to depict the relationship between each ecological factor towards overall accessibility to primary care services and how it varied across the study area. Data management and linear regression were conducted using STATA 15 (Stata Corp, College Station, TX, USA), while all spatial and network related analysis as well as GWR were conducted using ArcGIS Desktop 10.7 (ESRI, Redland, CA, USA).

## 3. Results

### 3.1. Spatial Accessibility to Primary Care in Rural Areas

The spatial patterns of E2SFCA scores are illustrated in Figure 2. At a glance, higher E2SFCA scores were found in areas that surround the urban centers. To be more specific, the central and southern region of rural areas of Selangor had better access, compared to the northern region. Detailed descriptive statistics for the E2SFCA scores across districts are displayed in Appendix A. From the descriptive statistics, the *Aspri* score for rural Selangor was slightly higher than the *Aspub* score (1.154 vs. 0.800), with an *Astot* score of 1.953. Indicating that for the public–private shares, about 59% of the *Astot* score was contributed by the *Aspri* score. In relation to predecessor version of E2SFCA (the 2SFCA), we also have conducted comparative assessment between the two methods in term of descriptive statistics and basic choropleth map as depicted in Appendix A. The 2SFCA method generally had significantly higher scores compared to the E2SFCA and smoother transition of score change across spatial area can be observed in E2SFCA, making the E2SFCA is more sensitive and favorable in identifying variation of score across space. Hot Spot Analysis (Figure 2) reveals the statistical significance of the spatial pattern depicted by the choropleth map. Cold spots for the *Aspub* score appeared to be along the coastal areas, as well as in several pockets of areas (HUS and HUL) inwards to the mountainous region in the trunk of Peninsular Malaysia. Hot spots for the *Aspub* score primarily appeared at the south region of the urban center of Selangor, and in the northern tip of HUS. For the *Aspri* score, hot spots were found near the urban center region of the state, while cold spots appeared in most of the coastal areas and central north area. The north region of Selangor appears to contain most of the cold spot areas, and it also can be summarized that low-access areas tend to be located further from the urban center.

Around 36.2%, 40.5%, and 44.5% of the total population of Selangor reside in the cold spot areas in relation to the *Aspub*, *Aspri*, and *Astot* scores, respectively. In terms of mean E2SFCA scores, the mean scores of the hot spot areas were generally about twice as high as average value, while the scores of the cold spot areas were about half of average value. Hypothetically, those population in hot spot areas had about four folds better opportunity to access the primary care compared to the population in the cold spot areas. Detailed results on population affected and differences in the E2SFCA scores across district were provided in the Appendix A. This study also explored the equality of the E2SFCA score distribution using the Gini coefficient (Appendix A). Across the whole studied area, the *Aspub* and *Astot* scores were fairly equally distributed, as compared to *Aspri* score. The discrepancies of access in term of the *Aspri* score were more prevalent in the north region.

### 3.2. Factors Associated to the Spatial Accessibility

The hierarchical linear regression (Table 2) shows that the E2SFCA scores were statistically differs across district and locality, indicating that the level of access significantly varied across different administrative districts and population resides the rural locality generally had higher E2SFCA scores (*Aspub* score (aOR = 0.27; 95% CI: 0.20,0.34); *Aspri* score (aOR = 0.16; 95% CI: 0.08,0.24), and *Astot* score (aOR = 0.41; 95% CI: 0.28,0.55) compared to population who resides in small urban locality. Shorter distances to nearest urban area were also associated with higher *Aspri* and *Astot* scores. Most studied demographic factors were associated with at least one of the E2SFCA scores, however no association of was found for household size, total dependency ratio and percentage of marginalized population with any of the E2SFCA score. Notable findings were that higher population density, road density, and proportion of Malay ethnic group in the EB were positively associated with all of the E2SFCA scores. Higher proportion of vulnerable population was positively associated, while female and old-age dependency ratio were negatively associated with the E2SFCA scores, particularly for *Aspri* and *Astot* scores. The geographical factors explained about 17.9%, 61.0% and 49.0% of total variation of the model (adjusted R^2^) for *Aspub*, *Aspri,* and *Astot* scores, respectively. Accounting both set of geographical and demographic factors slightly buffed the R^2^ values to 26.5%, 70.7%, and 59.3%, respectively.

GWR were then modelled based on the explanatory variables retained in the MLR Model II, particularly for *Astot* score. Initially there were nine (excluding interaction variables) explanatory variables retained however *district* and *strata* was omitted by default in the GWR model due to spatial autocorrelation as the variables were coded and aggregated based on geographical feature. Variable *female* was also then omitted due to multicollinearity (in GWR), leaving only six continuous explanatory variables in the GWR model. Slightly higher adjusted R^2^ (66.1% vs. 59.3%) and lower AIC values (3182.3 vs. 3423.5) were observed in the GWR model in comparison to the hierarchical MLR Model II. The GWR model is still comparable to the hierarchical MLR Model II, despite of have to omit three explanatory variables (*district*, *strata* and *female*), but with addition of able to depict how the coefficient varies spatially across study area, as the trade-off.

Results on MLR with just six explanatory variables (retained for GWR) were also showed in Table 3 for comparison. Although the direction of the association and the aOR values were similar (compared to MLR model II), lower adjusted R-square with only 45.9% of the total variation were explained by the six explanatory variables.

Distance to nearest urban area significantly affects the total spatial accessibility (*Astot* score) where regression coefficient increases gradually in diagonal direction from southeast to northwest of the state (Figure 3a). Indicating that the distance to nearest urban area were heavily affected the spatial accessibility in the northwest region of the state as compared to the south east region. Similar pattern was observed for population density (Figure 3b) and old-age dependency ratio (Figure 3e). Another notable finding is that road density has significant positive association with the spatial accessibility score, indicating areas with better road network has better access to primary care services and the GWR illustrated that stronger effect of road density towards spatial accessibility scores concentrated at the central region near the urban agglomeration, where the extensive road network is presence (Figure 3c). As for vulnerable population (Figure 3d) and Malay ethnicity (Figure 3f), the regression coefficient generally decreases gradually from southeast to northwest of the state.

## 4. Discussion

Malaysia has a unique mix of public and private healthcare delivery systems, with public clinics principally managed by the Ministry of Health (MOH) and private clinics largely funded through out-of-pocket payments or health insurance registered under MOH. Although public clinics are the main primary care service provider for both urban and rural populations catering to about 60% of total outpatient attendance [39,54], public primary care services mainly focus on serving rural and poor populations. On top of static brick-and-mortar clinics, there are also mobile services that attend to scheduled sites to serve populations of targeted remote areas [55]. For private clinics, these are mainly located in urban areas (Figure 1b), similar to what other studies have previously reported [8,9].

The rationale of calculating the spatial accessibility score to primary care services separately based on the sector (public or private) is to provide insight on how much the public and private sectors contribute to the Malaysian healthcare delivery system in a local context specifically in term of spatial access. Public clinics are accessible to all rural populations due to their affordable cost but this luxury may not be feasible for private clinics due to financial barriers. The total accessibility scores for primary care services, accounting for both public and private clinics, reflect the ideal accessibility level of the rural population when the financial barrier is lifted. State authorities introduced a healthcare subsidization scheme called “*Peduli Sihat*” to alleviate healthcare costs so poorer populations could utilize private clinics [56]. With the fact that the rural population had higher utilization and often congested the public facilities [43,54], the burden can be mitigated to the private sector if such subsidiary scheme can be fully benefited. Our study found that the rural population of Selangor actually had good spatial access to private clinics and most districts had equal or higher *Aspri* scores compared to their *Aspub* scores. Nonetheless, this study found that the private sector (*Aspri* scores) contributed to 59% of the total E2SFCA (*Astot*) scores, indicating that health policy related to public–private collaboration (e.g., the subsidization for poorer populations to utilize private clinics) is important for improving access to primary care services. Findings of this study also suggested that a good public–private partnership policy in primary care could remarkably improve access among the rural population and also represent rural areas that are located near to large urban agglomeration in any similar health care context globally.

During the calculation of E2SFCA scores, it was found that all rural EBs in Selangor could reach public clinics within the catchment size of 30 min (travel time), meaning that none of the EBs had *Aspub* and *Astot* scores of zero. Moreover, most EBs could reach any private clinic within the catchment size, and only 15 (1.1%) EBs had an *Aspri* score of zero. This indicates that the 30 min catchment size is sufficient and applicable for rural Selangor. However, if a larger scale study is to be conducted (macro or national level), using multiple catchment sizes will be crucial due to different threshold distances among the population. For example, urban populations should have a smaller catchment size due to greater population and service densities. However, for rural populations who must travel further as the health services are more dispersed, larger catchment sizes are required to avoid E2SFCA scores of zero [47]. As for our national context, it is possible to use the National Health and Morbidity Survey data to gauge optimal catchment size for each state or region for the variable catchment size approach, based on reported mean distance to seek primary care in each state or region. A similar approach in determining catchment sizes for specific regions or states could be applied for any other countries given the required variables were available in their national household surveys or census data. Additionally, we also have tested the differences between the 2SFCA and E2SFCA methods and our findings is in tandem with previous studies noting the inferiority of the 2SFCA over E2SFCA where 2SFCA estimates tend to overestimate less sensitive to depict spatial variations [11,21,28,32]. These findings further justify and the reasonableness of this study to adapting the E2SFCA method.

### 4.1. Spatial Pattern and Equality of E2SFCA Scores

This study exhibited that the discrepancies of access existed throughout the rural areas (Figure 2). The E2SFCA scores were higher near or surrounding the urban area in each district, especially surrounding the large urban agglomeration at the center of the state. These findings are similar to those commonly reported, where higher access primarily occurs near or surrounding large urban areas [10,11,12]. In our study, this may be due to the heavy concentration of private clinics in the large urban core at the center of the state along with the presence of an extensive road network, since the public clinics appear to have had a more dispersed distribution across both urban and rural areas throughout the state (Figure 1b), Further, by using the cut-off point of Gini coefficient < 0.4 as equal [57], it is proved that the public clinics were more equally distributed compared to the private clinics (Appendix A).

Meanwhile, areas with lower E2SFCA scores were more prominent in the northern region and along the coastal regions (Figure 2). The descriptive statistics (Appendix A) also revealed that three districts in the northern region (i.e., SBK, KUS, and HUS) had the lowest E2SFCA scores. Based on the 2019 Household Income and Expenditure Survey (HIES), those three districts were also among those with the lowest median household incomes [58] which could illustrate the relationship between lower accessibility and lower socioeconomic status of the population that has been commonly reported [19,35,36].

The cold spots were mostly located further from the urban center region of the state, particularly in the coastal areas (except at the center region). This is probably due to the imbalance between high population density and the relatively low number of clinics in along areas the coastal areas. Furthermore, the road network is not as extensive there as near the urban center region (as depicted in Figure 1b). The association between population density and the road network were then confirmed in the regression analysis. Although hot spot areas were mainly concentrated around the large urban center of the state, there were some hot spot areas located in the tip of the north region, which is due to the existence of public mobile services there. This finding could suggest that mobile services were relatively effective in alleviating access to primary care in the targeted area.

Discrepancies in access existed across the rural area, as depicted by the Hot Spot Analysis. Degree of inequality were then further investigated using Gini coefficient where the *Aspri* scores were the least equally distributed. The Gini also revealed that north region near to the mountainous region (HUS district) had the lowest degree of equality. This could be due to the geographic distributions of the population and clinics were spatially heterogeneous (Figure 1), causing the large variation in the E2SFCA scores. Despite those, the disparities in access to primary care in Selangor state were not severe. However, there is still room for improvement. Holistic assessment of accessibility could involve these two aspects: level of access (score) and equality. Rural areas in the northern region may had bigger accessibility issues, illustrated by relatively low accessibility scores, and even worse for areas with low degree of equality as well. Primary care services provisions could be concentrated in those areas by either adding new facilities, upgrading existing facilities or bringing services nearer to the population through mobile services.

### 4.2. Factors Associated to the E2SFCA Scores

This study explored and identified several ecological factors pertinent to accessibility of primary care services in rural areas, accounting for the type of provider: public and private. In spatial analysis, there are two model of regression analysis: (i) global and (ii) local [59]. In the global model, resulting statistics or parameter estimates are assumed to be constant across space and is the average coefficient across the study region. This is the common regression model. However, there are situations when the nature of relationship between variables are varied across space, referred as spatial nonstationary [60]. Local regression is an extension of the common regression framework equation to cater the spatial nonstationary, which is doable given that the data are drawn from geographical units. The details were extensively describe elsewhere [59,60,61]. In spatial analysis, both global and local regression is often performed to identify factors associated to the outcome of interest. In this study, the global regression model is the hierarchical multiple linear regression (MLR) model, used to estimate set of explanatory variables that are associated with the E2SFCA scores. The local regression model, geographically weighted regression (GWR) reinforced the findings by depicting the variation of the regression coefficient across studied area. However, only results on GWR for *Astot* score were shown (Figure 3) as the spatial pattern (results of the GWR) for all *Aspub*, *Aspri*, and *Astot* scores were similar.

The hierarchical MLR found that the E2SFCA scores were differ across districts, even after adjusting few explanation variables. Across administrative districts, three districts in the northern region (SBK, KUS, and HUS) had the lowest E2SFCA scores. This proved that the relationships are intrinsically different across space, thus the necessity of conducting local regression model. Any spatial variation of the coefficient can be depicted using the GWR and given the existence of spatial nonstationary relationships between the variables, GWR provides better estimates (better goodness-of-fit statistics) compared to the global regression model [61,62,63] and as demonstrated in this study.

Higher urbanization rate and higher spatial accessibility score to primary care services were commonly reported [19,24,30,64] due to the fact that health care facilities are more abundant at developed areas [47,65]. However, this study had demonstrated otherwise. Rural areas had slightly higher E2SFCA scores compared to the ‘small urban’ areas. It was reasonable that *Aspub* score in rural areas could be higher due to the government policy of developing the public primary care facilities by focusing more on serving the rural population [66,67]. As for the lower *Aspri* score in small urban areas, this could be due to higher competition and congestion in getting the health care services as small urban have higher population density as compared to in rural areas [29,68]. Interestingly this study actually had identified the existing ‘gap’ across the urban-rural spectrum in the studied area where the most vulnerable in term of accessibility appear to be the middle group (small urban), in contrast to the well documented positive association between degree of urbanization and level of accessibility to health care services [69,70]. With the continuous urban development and expansion, the government and local health authorities could spare some attention on the balance of population growth and health services development as current migration pattern of population inflow to the higher degree of urbanization areas [71,72] may eventually cause substantial population growth in that area and consequently leads to higher need to health care services.

Results on hierarchical MLR showed that E2SFCA scores decrease with the increasing distance to nearest urban area, particularly for *Aspri* and *Astot* scores. This is consistent with the literatures where population at closer proximity to urban areas often associated with higher spatial accessibility scores [12,21,28]. *Aspub* score however did not show significant association with distance to urban areas, indicating that access to public primary care were basically were not affected by the proximity to urban areas and it is in align with the MOH’s goals in providing equal access towards achieving the universal health coverage [66]. GWR analysis (Figure 3a) then revealed that the northwest region were the areas that most affected. In relation to the scarce urban areas in that region, the accessibility score is more sensitive to proximity to the urban areas implying that urban development (with wide-range of infrastructures including health services) is much more desirable there.

Hierarchical MLR indicates that the population density slightly influenced spatial accessibility, where higher populated area had better access, particularly for *Aspri* and *Astot* scores. This positive relationship was well documented [22,25,34,36,48,49,63,73]. GWR revealed that the population density had stronger effect on spatial access at the northern region, where the population density is much lower as compared to the other region in rural Selangor. This suggests for a rational planning of rural settlements to ensure the balance between service-to-population are beneficial to improve the accessibility in that region, especially since that region is currently undergoing rapid developments due to urban sprawl of the urban agglomeration and development of the northern corridor [74].

Areas with better road density signifies that area had better general infrastructure and build environment [75]. Lacking of those serves as barriers to accessing health care [28,33,36,63,76], which also demonstrated in the hierarchical MLR of this study. In the GWR, the effect of road density is significantly higher at the center region (Figure 3c), suggesting that the region had extensive road network and thus the population able to reach the facility with much ease. In contrast, the northern region may focus on improving the road infrastructure to improve accessibility of that region in general.

This study had interesting findings on the relationship between healthcare need (proxied via percentage of vulnerable population) and spatial accessibility score. Population with high healthcare need often reported to be negatively associated with the spatial accessibility score to healthcare [12,19,77]. Theoretically, the higher proportion of population with high healthcare need leads to reduced access due to higher utilization of services, competition and congestion in getting the [19,49]. This study however showed otherwise, particularly for the *Aspri* and *Astot* scores. The findings in this study could not really be explained by our current knowledge. Nonetheless, it also eventually indicates that the private clinics generally had good distribution across the health care need of population.

Socio-economic status (SES) is one of the most vital parameter of the population, where population with lower SES often reported to have lower accessibility scores to the health care services [12,19,21,35,36,73,77,78]. Due to limited data availability, dependency ratios were the only SES-related parameter that can be included in this study. Old-age dependency ratio showed a weak negative association to the accessibility scores. This does indirectly indicate that the more economically active the population (lower dependency ratio) is, the higher the accessibility scores, particularly for *Aspri* and *Astot* scores. Another crude and indirect indicator is the findings on districts with higher median household income (such as GBK, SPG, KLG, and HUL) [58] all have higher accessibility scores, while districts with lowest median income (such as SBK, HUS, and KUS) are the districts with lowest accessibility scores, located in the northern region. SES disadvantaged rural population is often presumed to be the most vulnerable group as it suffers from three main barriers in accessing health care: (i) geographical access; (ii) availability; and (iii) affordability [79]. Appropriate strategies related to alleviating the financial burden such as subsidization scheme or co-payment [80] (particularly to access the private clinic), or any strategies that reduced incurred cost related to transportation and travelling to reach the services would be very beneficial for the poor population.

Studies had reported that accessibility to health care services negatively associated with higher proportion of non-native or ethnic minorities [36,78,81]. This study found that racial composition associated with the accessibility scores where areas with higher proportion of native or ethnic majority (i.e., Malay) had better access. The lower accessibility among non-natives and ethnic minority involved in agricultural sector was also reported by studies in China [36,81], which in this case could be similar as in our local context where a higher proportion of the non-Malay (the non-native ethnic group; especially Indian and non-citizen) population resides in deep rural area and are involved with the rural agricultural jobs (especially palm oil) [82,83]. This idea was also supported by the findings in GWR (Figure 3f) where the northern district contains the hinterland areas for paddy and agricultural conservation zone [46]. Although the negative association between accessibility score and proportion of marginalized population such as aborigine and immigrant were reported [12,35,49,77], this study cannot confirm the association. This could possible due to low proportion of those population in our study area (6.5%), making it difficult to reach statistical significance [84] in multivariable (MLR and GWR) analysis, despite showing a significant association in univariate analysis.

### 4.3. Summary

Considering the dual public-private primary care providers, this paper comprehensively depicts the spatial accessibility of primary care in the context of rural Selangor as well as identifying the associated factors. These findings could also represent and applicable for rural areas that are located near to any large urban agglomeration in Malaysia. To produce an accurate results for the spatial accessibility (E2SFCA scores calculation), this study: (i) used smallest-area aggregation data (i.e., enumeration block, EB), to produce homogenous areal units, as well as; (ii) including data on population and health facilities from adjacent states to eliminate the edge effect [31]. Then, this study further modifies the original formula by incorporating the health need (*HN*) and clinic availability (*CA*). The conceptualization of spatial accessibility in this study accounts for the *HN* of the population and the availability of each clinic. Higher *HN* for a population implies a higher utilization of primary care services, causing reduced access due to higher competition and congestion. This is consistent with the National Health and Morbidity Survey findings that vulnerable populations (i.e., toddlers, elders, and women aged 15–45) had a higher frequency of annual outpatient visits [54,85]. In regards to the *CA*, Khakh et al. (2019) suggested that service hours and working days of clinics can be considered to further improve the measure of access [86]. Hence, this study incorporated the *CA* into the E2SFCA calculation. Higher *CA* means that the population had a wider window of opportunity to obtain healthcare services, given that the clinics had longer operating hours. A future study could be conducted to explore how accessibility differs between office hours (day), and after office hours (night), as temporal differences of accessibility could exist depending on the premise’s schedule [87]. Next, this study deliberately explored ecological factors that influence the level of spatial accessibility and depicted the varying effects of each explanatory variables on the level of accessibility over space, which would be very beneficial for the government (or any authorities) to propose specific improvement measures according to the local conditions. Nonetheless, this study has demonstrated the resourcefulness and practicability of data integration, served as a proof of concept of the extended potential use benefiting from multiple databases with data routinely collected by government agencies with few tweaks in the current E2SFCA method to further improve the conceptualization of healthcare access and suit to our local context. Continuous monitoring of the performance of primary healthcare accessibility could be feasibly conducted.

### 4.4. Limitations

There were some limitations of this study. Population data were based on Census 2010, which is the only and most recent available data during the study commenced. It was estimated that Selangor had an average population growth of 1.9% annually for the last 10 years [88], therefore the calculated E2SFCA score in this study is underestimated due to the increase of population. Based on the recent Census 2020 report [89], the rural population in Selangor has shrank in 2020 as compared to 2010 due to opening of new settlements or cities [90], and rapid urban expansion [91] throughout the period (the urbanization in Selangor state has increased to 95.8% in the year 2020, as compared to 91.3% in 2010) and different rate of urbanization between districts were also observed. Detailed changes of urbanization rate and rural population were described in Appendix A. Despite the reduction of the rural population, it was only due to the changes of the stratum category, or in other words, some of the rural EBs were no longer rural at current years. Therefore, the geographical mapped E2SFCA scores shall still applicable to depict the spatial access, however still need to be interpreted with cautious, considering the mentioned dynamic change of the population and the effect of underestimation is different between district due to the different population growth (as tabulated in Appendix A). Therefore, future works could be conducted using the most recent population census data and by addressing those limitations that revolve around data availability in order to produce better and contemporary estimates. The number of doctors used to weight the primary care service capacity at each facility was assumed. Another limitation was that the private clinics’ operating hours were based on binary classification of 24-h statuses. These two limitations are due to data unavailability during this study commenced. Nonetheless, this study had demonstrated the practical use of such estimation that can adopted by fellow researchers as data availability could be the main challenge for some countries in which developing countries often lacking.

## 5. Conclusions

In conclusion, this study has contributed to a better conceptualization of accessibility and the E2SFCA method by incorporating the health need of populations and facility availability into the original formula. This method can be applied to other similar healthcare settings and contexts. The practical use of routinely collected or available data and alternative approaches in handling the issues of data unavailability, as well as adjustment of certain parameter in the E2SFCA formula using national survey findings, surely contribute to the body of knowledge related to the methodological issues. This study then performed a comprehensive assessment on the spatial accessibility of primary care for rural areas in terms of spatial pattern, identification of low and high accessibility areas, quantification of the population affected, and the degree of equality of the spatial accessibility. On top the standard linear regression in identifying the associated factors, the application of GWR has enrich the findings by exploring how the factors varied across the studied area—this would be critical to aid policymakers and health authorities in identifying areas that need attention for a more specific and localized improvement plan considering the limited national resources.

## Figures and Tables

**Figure 1 ijerph-20-03147-f001:**
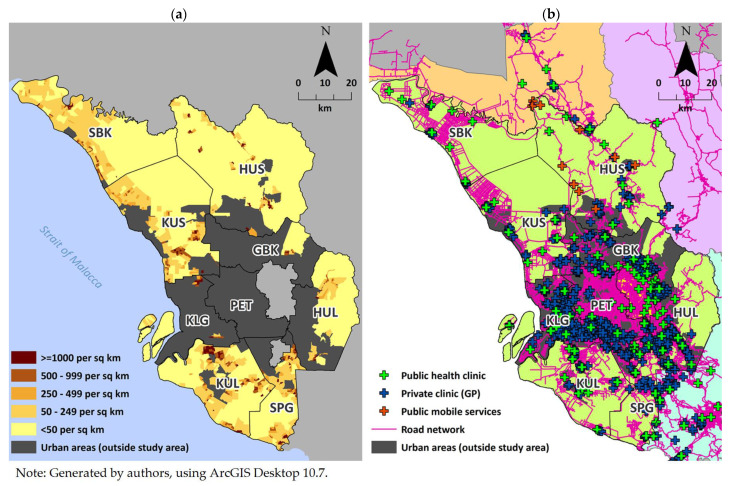
Study area and districts from adjacent district/states. (**a**) Population density (person per square kilometer) of rural Selangor. (**b**) Primary care facility distribution (by clinic type) and road network.

**Figure 2 ijerph-20-03147-f002:**
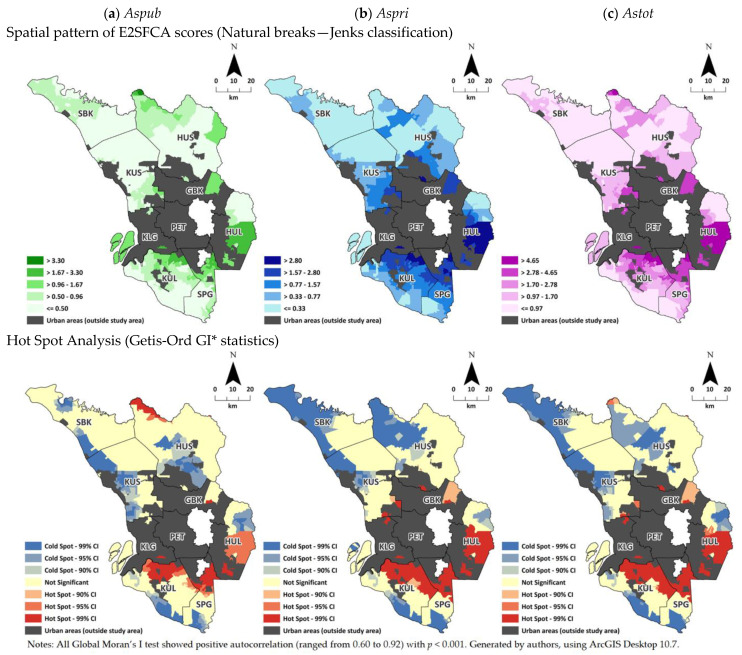
Spatial pattern of E2SFCA scores distribution and mapping of Hot Spot Analysis of the E2SFCA scores (**a**) *Aspub* score, (**b**) *Aspri* score, and (**c**) *Astot* score.

**Figure 3 ijerph-20-03147-f003:**
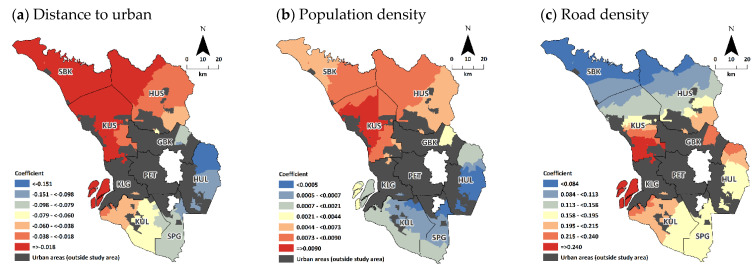
Regression coefficient of each explanatory variables (**a**–**f**) and standardized residuals (**g**) of the GWR on *Astot* score.

**Table 1 ijerph-20-03147-t001:** Descriptive information of study population characteristics.

Variables, Unit	Definition	Mean ± SD	
District			
Gombak	Residents of Gombak (GBK) district	0.01	±0.10
Hulu Langat	Residents of Hulu Langat (HUL) district	0.05	±0.22
Hulu Selangor	Residents of Hulu Selangor (HUS) district	0.15	±0.36
Klang	Residents of Klang (KLG) district	0.04	±0.18
Kuala Langat	Residents of Kuala Langat (KUL) district	0.26	±0.44
Kuala Selangor	Residents of Kuala Selangor (KUS) district	0.18	±0.38
Sabak Bernam	Residents of Sabak Bernam (SBK) district	0.15	±0.36
Sepang	Residents of Sepang (SPG) district	0.17	±0.37
Locality/stratum			
Small Urban	EB with 1000–9999 person	0.22	±0.41
Rural	EB with <1000 persons and non-gazetted areas	0.78	±0.41
Distance to nearest urban area, minutes	Based on travelling via road network using motorized vehicle as mode of transport	8.8	±8.7
Population density, person per acre	Total number of persons divided by total land area of the EB	10.7	±25.1
Road density, km road per km^2^ of land area	Total length of road divided by total land area of the EB	3.4	±2.5
Household size	Number of household member	4.3	±0.58
Female, %	Proportion of female population in the EB	0.48	±0.06
Vulnerable population, %	Proportion of vulnerable population (toddler aged <5, elder aged >64 and female aged 15–45) in the EB	0.37	±0.05
Total dependency ratio (TDR)	Ratio of dependent population in relation to working age population (aged 15–64) in the EB. TDR is the sum of old-age and youth dependency ratios.	0.57	±0.18
Old-age dependency ratio	Old-age dependent are population aged >64, in relation to working age group (aged 15–64) in the EB	0.09	±0.07
Youth dependency ratio	Youth dependent are population aged <15, in relation to working age group (aged 15–64) in the EB	0.48	±0.16
Malay, %	Proportion of Malay ethnic population in the EB	0.72	±0.32
Marginalized, %	Proportion of marginalized population (aborigine and non-citizen) in the EB	0.07	±0.13

**Table 2 ijerph-20-03147-t002:** Association of geographical and demographic factors with E2SFCA scores.

	Aspub						Aspri						Astot					
	Crude β		Adj. β				Crude β		Adj. β				Crude β		Adj. β			
			Model I		Model II				Model I		Model II				Model I		Model II	
Geographical factors																		
District																		
Gombak (GBK)	0.764	***	0.732	***	0.736	***	2.475	***	1.703	***	1.482	***	3.239	***	2.326	***	2.174	***
Hulu Langat (HUL)	0.133		0.259	**	0.268	***	1.263	***	0.810	***	0.766	***	1.395	***	0.992	***	1.026	***
Hulu Selangor (HUS)	0.153	**	0.219	***	0.253	***	0.597	***	0.377	***	0.356	***	0.750	***	0.557	***	0.607	***
Klang (KLG)	0.665	***	0.633	***	0.547	***	1.572	***	1.634	***	1.336	***	2.237	***	2.280	***	1.881	***
Kuala Langat (KUL)	0.267	***	0.271	***	0.211	***	1.243	***	0.849	***	0.648	***	1.509	***	1.063	***	0.853	***
Kuala Selangor (KUS)	−0.130	**	−0.119	**	−0.109	*	0.483	***	−0.039		0.055		0.353	***	−0.235	*	−0.036	
Sabak Bernam (SBK)	Ref		Ref		Ref		Ref		Ref		Ref		Ref		Ref		Ref	
Sepang (SPG)	0.499	***	0.547	***	0.415	***	1.535	***	1.114	***	0.757	***	2.034	***	1.596	***	1.168	***
Locality/stratum																		
Small urban	Ref		Ref		Ref		Ref		Ref		Ref		Ref		Ref		Ref	
Rural	0.223	***	0.266	***	0.270	***	0.080		0.172	***	0.160	***	0.303	**	0.439	***	0.414	***
Distance to nearest urban area	−0.007	***	-		-		−0.059	***	−0.055	***	−0.038	***	−0.067	***	−0.063	***	−0.036	***
Demographic factors																		
Population density	0.002	**	-		-		0.010	***	-		0.003	***	0.012	***	-		0.003	*
Road density	0.091	***	-		0.071	***	0.228	***			0.104	***	0.319	***	-		0.178	***
Household size	0.038		-		-		0.188	***	-		-		0.226	***	-		-	
Female, %	−0.233		-		-		−1.692	***	-		−1.876	***	−1.924	**	-		−2.066	***
Vulnerable population, %	0.383		-		-		1.195	**	-		1.918	***	1.578	**	-		2.152	***
Total dependency ratio	−0.002		-		-		−0.005	***	-		-		−0.007	**	-		-	
Old-age dependency ratio	−0.012	***	-		-		−0.053	***	-		−0.013	***	−0.065	***	-		−0.012	***
Youth-age dependency ratio	0.001		-		−0.002	*	0.004	**	-		-		0.005	*	-		-	
Malay ethnic, %	0.162	**	-		0.177	***	0.201	**	-		0.218	***	0.363	**	-		0.392	***
Marginalised population, %	0.249	*	-		-		0.894	***	-		0.012		1.143	***	-		0.137	
Malay * marginalised	-		-		-		-		-		1.565	***	-		-		1.967	***
Statistics																		
Constant/intercept	-		0.388		0.151		-		0.995		0.700		-		1.504		0.723	
Adjusted R-square (R^2^)	-		0.179		0.265		-		0.610		0.707		-		0.490		0.593	
AIC	-		2129.8		1980.3		-		2374.8		1990.0		-		3731.1		3423.5	

Notes: [1] Simplified results showing β-coefficient and significance level, indicated by * 0.01 <= *p* < 0.05, ** 0.001 <= *p* < 0.01, *** *p* < 0.001. Full table showing 95% CI and *p*-values are provided as in Appendix A. [2] Model I = controlled for geographical factors; Model II = controlled for geographical factors and demographic profiles (full model). [3] Abbreviations: Ref = reference group; dash (-) = not included in the model, or not applicable; AIC = Akaike’s information criterion.

**Table 3 ijerph-20-03147-t003:** Multiple Linear Regression (MLR), factors associated to Total E2SFCA scores (*Astot*).

	Adj. β	SE	95% CI		*p*-Value
			Lower	Upper	
Explanatory variables in the model [1]					
Distance to nearest urban area, minutes [2]	−0.030	0.003	−0.037	−0.024	<0.001
Population density (person per acre)	0.003	0.001	0.001	0.005	0.003
Road density (km road per km^2^ of land area)	0.227	0.013	0.202	0.252	<0.001
Vulnerable population (elder and toddler), % [3]	1.549	0.459	0.648	2.450	0.001
Old-age dependency ratio [4]	−0.031	0.004	−0.039	−0.023	<0.001
Malay, %	0.102	0.083	−0.061	0.265	0.221
Statistics					
Constant/intercept	0.932	0.179	0.581	1.282	<0.001
Adjusted R-square	0.459				
Akaike Information Criterion (AIC)	3796.0				

Notes: [1] Only explanatory variables included for Geographical Weight Regression (GWR) were retained. [2] Travel distance to nearest urban area in minute via road network, using car/motorized vehicle. [3] Vulnerable population = toddler aged ≤5, elder aged ≥65, female aged 15–45. [4] Old dependency ratio = ratio of old population (aged ≥65) in relation to the working age (15–64 years).

## Data Availability

Restrictions apply to the availability of these data. Population data was obtained from Department of Statistics Malaysia (DOSM); road network and geographical boundary was obtained from National Geospatial Centre Malaysia and are available from the authors upon reasonable request and with the permission of the respective data custodian. Those data were not publicly available due to data sharing agreement. Data on primary care facilities prepared by Ministry of Health Malaysia are publicly available, obtainable from https://www.data.gov.my/ (accessed on 6 May 2019) and https://medicalprac.moh.gov.my/statistik/ (accessed on 26 March 2019).

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
