# Peer review of "Spatial Accessibility of Primary Care in the Dual Public–Private Health System in Rural Areas, Malaysia"

_ijerph, 2023, doi:10.3390/ijerph20043147_

Round 1
Reviewer 1 Report
(1) Part 2.3 proposes to first list the source, calculation formula and limitations of the two-step floating catchment area (2SFCA) method, which leads to the improved content and calculation formula of E2SFCA. Finally, the comparison summarizes the advantages of E2SFCA over 2SFCA.
(2) The Result section lacks quantitative comparison arguments to illustrate the more accurate results of E2SFCA. It is suggested that the calculation results of 2SFCA be added to 3.1, and a new subsection is added to quantitatively compare the results of 2SFCA and E2SFCA to fully discuss the reasonableness of the calculation results of E2SFCA.
Author Response
Reviewer #1:
Point 1: Part 2.3 proposes to first list the source, calculation formula and limitations of the two-step floating catchment area (2SFCA) method, which leads to the improved content and calculation formula of E2SFCA. Finally, the comparison summarizes the advantages of E2SFCA over 2SFCA.
Response 1: Thank you for suggestion on improving the flow and methodology of the manuscript. However, based on our knowledge while conducting the literature search for this study, we have some across many studies comparing over the two methods (2SFCA and E2SFCA) and the advantages of E2SFCA over 2SFCA were well documented (Delamater 2013; Luo & Qi, 2009; Wan et al., 2012) where 2SFCA often tend to overestimate the score as it assume equal access within the catchment and does not considered the distance decay concept (Luo & Qi, 2009; Hu et al., 2013; Donohoe et al., 2016). Some studies also depicted the E2SFCA had smoother transition of score change across spatial area and more sensitive in identifying clusters (Lin et al., 2016; Luo & Qi 2009). Although in some cases, there were also studies that found the 2SFCA and E2SFCA yield similar results, however most literatures suggest that the superiority of the E2SFCA over 2SFCA. Thus, we decided to directly adapt the E2SFCA method without showing how 2SFCA performed in this study.
Taking your point, we have added the points above in introduction section (Page 2, line 59–62) and also added the results comparing the 2SFCA method and E2SFCA method as provided in the supplementary material as comparing method is initially not the focus of your study. Nonetheless, it still be noteworthy to add the suggested additional analysis (as stated per your 2nd point), in supplementary materials. (Only if you can accept/agree to this, if not, we will put in the results in main manuscript as per your initial suggestion).
Below is the list of reference as per cited in the above paragraph, to support our justification:
- Lin, B. C.; Chen, C. W.; Chen, C. C.; Kuo, C. L.; Fan, I. chun; Ho, C. K.; Liu, I. C.; Chan, T. C. Spatial Decision on Allocating Automated External Defibrillators (AED) in Communities by Multi-Criterion Two-Step Floating Catchment Area (MC2SFCA). Int J Health Geogr, 2016, 15 (1), 1–14. https://doi.org/10.1186/s12942-016-0046-8.
- Delamater, P. L. Spatial Accessibility in Suboptimally Configured Health Care Systems: A Modified Two-Step Floating Catchment Area (M2SFCA) Metric. Heal Place, 2013, 24, 30–43. https://doi.org/10.1016/j.healthplace.2013.07.012.
- Luo, W.; Qi, Y. An Enhanced Two-Step Floating Catchment Area (E2SFCA) Method for Measuring Spatial Accessibility to Primary Care Physicians. Health Place, 2009, 15 (4), 1100–1107. https://doi.org/10.1016/j.healthplace.2009.06.002.
- Dewulf, B.; Neutens, T.; De Weerdt, Y.; Van De Weghe, N. Accessibility to Primary Health Care in Belgium: An Evaluation of Policies Awarding Financial Assistance in Shortage Areas. BMC Fam Pract, 2013, 14 (1), 1–13. https://doi.org/10.1186/1471-2296-14-122.
- Hu, R.; Dong, S.; Zhao, Y.; Hu, H.; Li, Z. Assessing Potential Spatial Accessibility of Health Services in Rural China : A Case Study of Donghai County. Int J Equity Health, 2013, 12 (1), 1–11.
- Donohoe, J.; Marshall, V.; Tan, X.; Camacho, F. T.; Anderson, R. T.; Balkrishnan, R. Spatial Access to Primary Care Providers in Appalachia: Evaluating Current Methodology. J Prim Care Community Heal, 2016, 7 (3), 149–158. https://doi.org/10.1177/2150131916632554.
- Wan, N.; Zou, B.; Sternberg, T. A Three-Step Floating Catchment Area Method for Analyzing Spatial Access to Health Services. Int J Geogr Inf Sci, 2012, 26 (6), 1073–1089. https://doi.org/10.1080/13658816.2011.624987.
Point 2: The Result section lacks quantitative comparison arguments to illustrate the more accurate results of E2SFCA. It is suggested that the calculation results of2SFCA be added to 3.1, and a new subsection is added to quantitatively compare the results of 2SFCA and E2SFCA to fully discuss the reasonableness of the calculation results of E2SFCA.
Response 2: Thank you for comments as to further improve the comprehensiveness of the results as well as the strengthening the justification of this study to use E2SFCA by demonstrating the quantitative comparison of the 2SFCA and E2SFCA. We have added the comparison as in supplementary materials (Figure S2). We also added some sentence in regards to this matter (refer section Results 3.1, first paragraph, page 7, line 273–279), and briefly mentioned in the Discussion section (before section 4.1, page 14, last paragraph).

Reviewer 2 Report
This article is focusing on spatial accessibility of primary care in the dual public–private health system in rural areas in Malaysia. The article is using Enhance 2-Step Floating Catchment Area (E2SFCA) method, Hot spot Analysis and Hierarchical multiple linear regression and geographical weight regression to analyze systematically the factors which affect the spatial accessibility of primary care accordingly. This article is developed in a full framework from the literature discussions to the empirical research findings, providing an updated understanding about the primary care development in Malaysia. However, based on the population data of Census 2010, the data is a bit too old to show and analyze the current situation and issues. Thus, the conclusions and policy implications about the primary care development in Malaysia seems to be updated in order to enhance the research contributions.
Author Response
Reviewer #2:
This article is focusing on spatial accessibility of primary care in the dual public–private health system in rural areas in Malaysia. The article is using Enhance 2-Step Floating Catchment Area (E2SFCA) method, Hot spot Analysis and Hierarchical multiple linear regression and geographical weight regression to analyze systematically the factors which affect the spatial accessibility of primary care accordingly. This article is developed in a full framework from the literature discussions to the empirical research findings, providing an updated understanding about the primary care development in Malaysia.
Point 1: However, based on the population data of Census 2010, the data is a bit too old to show and analyze the current situation and issues. Thus, the conclusions and policy implications about the primary care development in Malaysia seems to be updated in order to enhance the research contributions.
Response 1: Thank you for your comments on raising the concern of validity of the results on the policy implication upon the findings, which is similar to Review #3’s comment. We agree and aware of the limitation of using the Census 2010 as the data is quite old. However, that is the only census data available during the study commenced and next census was conducted in 2020 and slightly delayed to 2021 due to COVID-19 pandemic. It also not possible for research team to obtain the recent 2020 data as well; due to ethical approval only up to using 2010 data and requesting the recent data may further delay the research and publication. Although Department of Statistics Malaysia has produced intercensal population estimates, however it only aggregated up to state level (while this study requires finest data aggregated at enumeration block level for triangulation and analysis). Based on recent census 2020, the total population of rural Selangor has reduced by 37.2% of what reported in 2010 (DOSM 2022), but the reduction is only due to the urban expansion and some the rural EBs no longer “rural” at current year. In fact, the total population in Selangor has increased, making the calculated E2SFCA score in this study is underestimated due to increase of population. Nonetheless, the mapped E2SFCA score in this study is still applicable, but may need to be interpreted with cautious as it may or might not reflect “current situation” due to the aforementioned underestimation effect.
We take note of your concern, we have moved 2010 data issue as the first limitation, and the implication of the underestimation (section 4.4, page 17). We also have added supplementary file (Table S4) to depict the changes of population across Selangor and within districts. We also had slightly revised the research question from “what is the current situation …” changed to emphasising that this manuscript is focusing more on showing the approach and assessment of spatial accessibility priori to Malaysia’s experience and local context, rather than the actual/current level of spatial accessibility per se. This study also served as proof of concept and demonstrated the resourcefulness and the extended potential use of the secondary data that were readily available and routinely collected by various government agencies, hoping that these findings will reach them and further nurture the data sharing and integrating practices, not just for academia purposes.
- DOSM, 2012. Current Population Estimates, Administrative District, 2022: Selangor. Department of Statistics Malaysia. Putrajaya. ISBN: 978-967-253-694-9. https://www.dosm.gov.my

Reviewer 3 Report
this is an interesting study using geographical level analysis.
1. this study using several different dataset with different time point. the data about district used 2010 data but the data related with clinics used 2017 data. please give explanation whether there is change in district level data between 2010-2017? is there no new update related with census data other than 2010 data?
2. please include the model result from linear model in the manuscript for comparison
3. is there any specific reason for removing district, strata and female on GIS analysis?
Author Response
Point 1: This is an interesting study using geographical level analysis. This study using several different datasets with different time point. the data about district used 2010 data but the data related with clinics used 2017 data. please give explanation whether there is change in district level data between2010-2017? is there no new update related with census data other than 2010 data?
Response 1: Thank you for your comments on raising the concern of validity of the results on the policy implication upon the findings, which is similar to Review #2’s comment. We agree and aware of the limitation of using the Census 2010 as the data is quite old. However, that is the only census data available during the study commenced and next census was conducted in 2020 and slightly delayed to 2021 due to COVID-19 pandemic. It also not possible for research team to obtain the recent 2020 data as well; due to ethical approval only up to using 2010 data and requesting the recent data may further delay the research and publication. Although Department of Statistics Malaysia has produced intercensal population estimates, however it only aggregated up to state level (while this study requires finest data aggregated at enumeration block level for triangulation and analysis). Based on recent census 2020, the total population of rural Selangor has reduced by 37.2% of what reported in 2010 (DOSM 2022), but the reduction is only due to the urban expansion and some the rural EBs no longer “rural” at current year. In fact, the total population in Selangor has increased, making the calculated E2SFCA score in this study is underestimated due to increase of population. Nonetheless, the mapped E2SFCA score in this study is still applicable, but may need to be interpreted with cautious as it may or might not reflect “current situation” due to the aforementioned underestimation effect.
We take note of your concern, we have moved 2010 data issue as the first limitation, and the implication of the underestimation (section 4.4, page 17). We also have added supplementary file (Table S4) to depict the changes of population across Selangor and within districts. We also had slightly revised the research question from “what is the current situation …” and emphasising that this manuscript is focusing more on showing the approach and assessment of spatial accessibility priori to Malaysia’s experience and local context, rather than the actual/current level of spatial accessibility per se. This study also served as proof of concept and demonstrated the resourcefulness and the extended potential use of the secondary data that were readily available and routinely collected by various government agencies, hoping that these findings will reach them and further nurture the data sharing and integrating practices, not just for academia purposes.
- DOSM, 2012. Current Population Estimates, Administrative District, 2022: Selangor. Department of Statistics Malaysia. Putrajaya. ISBN: 978-967-253-694-9. https://www.dosm.gov.my
Point 2: Please include the model result from linear model in the manuscript for comparison
Response 2: Thank you for the comments in further improving and enriching the results section. We have added the multiple linear model in relation to the variables that retained for GWR as suggested in Results section 3.2, page 11, Table 3. Relevant write-up was added in Results section 3.2, page 9, last paragraph.
Point 3: Is there any specific reason for removing district, strata and female on GIS analysis?
Response 3: The district, strata and female were removed due to spatial autocorrelation issue in the Geographical Weight Regression (GWR). In case of variable district and strata, are the variables that are coded and aggregated based on geographical feature. If the district and strata values were mapped across study area, a cluster of values (of district and strata) can be seen. For example, all EBs in same district / strata will have same value (due to numerical coding), thus autocorrelation occurs as the EB and its neighbouring/nearby EB has same values making the variables (district and strata) ineligible for GWR analysis by default. Female variable is slightly different, the autocorrelation occurs by chance (neighbouring EBs tend to have similar values across studied area, which also showed in Table 1; the average % of female was 48% with SD of ±6).
We have revised some sentences related to the reason of removing those variables in Results section 3.2, page 9, second last paragraph.

Round 2
Reviewer 1 Report
The modified manuscript is in line with expectations.